# Multifunctional TiO_2_ Nanotube-Matrix Composites with Enhanced Photocatalysis and Lithium-Ion Storage Performances

**DOI:** 10.3390/ma16072716

**Published:** 2023-03-29

**Authors:** Mengmeng Zhang, Hui Li, Chunrui Wang

**Affiliations:** Shanghai Institute of Intelligent Electronics and Systems, College of Science, Donghua University, Shanghai 201620, China

**Keywords:** double-layer honeycomb structure, multifunctional composite, photocatalytic degradation, lithium-ion battery, high specific capacity

## Abstract

As a multifunctional material, TiO_2_ shows excellent performance in catalytic degradation and lithium-ion storage. However, high electron-hole pair recombination, poor conductivity, and low theoretical capacity severely limit the practical application of TiO_2_. Herein, TiO_2_ nanotube (TiO_2_ NT) with a novel double-layer honeycomb structure were prepared by two-step electrochemical anodization. Honeycombed TiO_2_ NT arrays possess clean top surfaces and a long-range ordering, which greatly facilitates the preparation of high-performance binary and ternary materials. A binary TiO_2_ nanotube@Au nanoparticle (TiO_2_ NT@Au NP) composite accompanied by appropriately concentrated and uniformly distributed gold particles was prepared in this work. Interestingly, the TiO_2_ nanotube@Au nanoparticle (TiO_2_ NT@Au NP) composites not only showed the excellent catalytic degradation effect of methylene blue, but also demonstrated large lithium-ion storage capacity (310.6 μAh cm^−2^, 1.6 times of pristine TiO_2_ NT). Based on the realization of the controllable fabrication of binary TiO_2_ nanotube@MoS_2_ nanosheet (TiO_2_ NT@MoS_2_ NS) composite, ternary TiO_2_ nanotube@MoS_2_ nanosheet@Au nanoparticle (TiO_2_ NT@MoS_2_ NS@Au NP) composite with abundant defects and highly ordered structure was also innovatively designed and fabricated. As expected, the TiO_2_ NT@MoS_2_ NS@Au NP anode exhibits extremely high initial discharge specific capacity (487.4 μAh cm^−2^, 2.6 times of pristine TiO_2_ NT) and excellent capacity retention (81.0%).

## 1. Introduction

With the booming development of science and technology, the environment and energy issues attracted more and more attention. As a multifunctional semiconductor material, TiO_2_ can be used as catalyst and anode in photocatalytic degradation and lithium batteries [1,2,3]. TiO_2_ has the advantages of being non-toxic, safety, and stability, so it is widely pursued by researchers [4]. The construction of various nanostructures (e.g., nanoparticle, nanowire, nanorod, and nanotube) brought the advantage of large specific surface area to TiO_2_ nanomaterials, which is more conducive to the improvement of TiO_2_ nanomaterials’ catalytic and electrochemical properties [3,5,6]. In particular, the TiO_2_ nanotube prepared by electrochemical anodizing has a self-organized morphology, and the tubular structure shortens the transmission path of electrons and ions, greatly increases the specific surface area, making the TiO_2_ nanotube an alternative catalyst and anode [3,7]. However, high electron-hole pair recombination, poor conductivity, and inferior theoretical capacity (335 mAh g^−1^) limit the practical application of TiO_2_ nanotubes in catalysis and lithium-ion battery [8,9,10].

As a widely known metal, Au has the characteristics of inactivity and excellent conductivity. While the metal/semiconductor (Au/TiO_2_) junction appears, Au as the separation center can decrease the recombination of electron-hole pairs in the TiO_2_ and overcome shortcoming of high electron-hole pair recombination of TiO_2_ semiconductor materials, leading to the enhanced degradation ability of organic pollutants [11,12,13]. On the other hand, Au is a good conductive additive, which can significantly increase the conductivity of TiO_2_ anode. The addition of Au will increase the transfer rate of ions and electrons in TiO_2_ nanotubes, but the improvement of lithium capacity is very limited [14,15,16]. Thus, it is still a challenge to further improve the TiO_2_ nanotube-matrix composite electrode’s capacity. Molybdenum sulfide (MoS_2_), as a typical two-dimensional material, is considered as an ideal candidate anode for lithium batteries because of its large theoretical specific capacity (670 mAh g^−1^) and high safety [17,18,19]. However, the large volume expansion of MoS_2_ in the process of charging/discharging leads to the large capacity instability of lithium batteries [20,21]. Considering the high stability of TiO_2_, many researchers combined TiO_2_ with MoS_2_, which combined the advantages of the two components to prepare high-performance composite lithium battery anode [17,22]. In previous studies, TiO_2_@MoS_2_ composites were mostly prepared using one-step anodization TiO_2_ nanotubes as a matrix. However, such traditional nanotube’s disordered top surface and distinct nanotube length lead to random accumulation of MoS_2_ in the composite [23,24]. Therefore, developing a clean top surface and long-range ordered TiO_2_ nanotube matrix is very important for the preparation and performance of composite materials.

In this work, the TiO_2_ nanotube (TiO_2_ NT) with a double-layer honeycomb structure was prepared by two-step electrochemical anodization. Fancifully, honeycombed TiO_2_ NT differ from traditional one-step oxidation nanotubes in that they have a clean top surface and a long-range ordering. In addition, honeycombed TiO_2_ NT arrays have porous properties and large specific surface areas, which contribute to the preparation of high-performance composites by combining them with Au nanoparticles and MoS_2_ nanosheets. High performance multifunctional TiO_2_ nanotube@Au nanoparticle (TiO_2_ NT@Au NP) composites accompanied by uniformly distributed Au nanoparticles were successfully prepared in this work. Excitedly, the TiO_2_ NT@Au NP composite shows excellent catalytic degradation effect, due to the existence of Au particles with appropriate concentration in composite. In addition, while the TiO_2_ NT@Au NP composite is used as a lithium-ion battery’s anode, it also demonstrates large initial specific capacity (310.6 μAh cm^−2^, 1.6 times of pristine TiO_2_ NT) and high initial coulomb efficiency (76.5%). Furthermore, ternary TiO_2_ nanotube@MoS_2_ nanosheet@Au nanoparticle (TiO_2_ NT@MoS_2_ NS@Au NP) anode was also successfully designed and synthesized in this work. Compared with traditional binary materials, such ternary material possesses abundant defects and highly ordered structure, improving the kinetics properties and lithium storage capacity of TiO_2_ NT@MoS_2_ NS@Au NP composite. Thus, the as-designed TiO_2_ NT@MoS_2_ NS@Au NP anode exhibited remarkable cycle stability (81.0% capacity retention) and large lithium-ion capacity (487.4 μAh cm^−2^, 2.6 times of pristine TiO_2_ NT).

## 2. Materials and Methods

### 2.1. Synthesis of TiO_2_ Nanotube (TiO_2_ NT)

Based on previous work, the electrochemical anodization method was used to prepare TiO_2_ nanotube array [2]. First anodizing process was carried out at 60 V for 60 min, and then the TiO_2_ NT array was peeled off by sonication. Subsequently, the second electrochemical anodization was implemented for 20 min (60 min) at a constant voltage of 60 V. Then, the secondary oxidized TiO_2_ nanotube was annealed at 450 °C in air atmosphere for 2 h to obtain the as-annealed secondary oxidized TiO_2_ nanotubes, named TiO_2_ NT, which was used as a matrix to prepare subsequent composites.

### 2.2. Synthesis of TiO_2_ Nanotube@Au Nanoparticle (TiO_2_ NT@Au NP) and TiO_2_ Nanotube@MoS_2_ Nanosheet (TiO_2_ NT@MoS_2_ NS)

Next, 1 mL of sodium citrate aqueous solution (1.5 wt%) and 1 mL of 1.2 mM polyvinylpyrrolidone (PVP, 58,000 g/mol) aqueous solution were added to 43 mL ultra-pure water. The mixed solution was transferred into a round bottom flask; meanwhile, the as-prepared TiO_2_ NT was suspended in the mixed solution. The mixed aqueous solution was heated in the oil bath at 115 °C until it boiled, and then we added 5 mL of 2.5 mM tetrachloroauric acid aqueous solution into the reactor. Under magnetic stirring, the reaction duration was 60 min. Then, we took out the titanium foil attached with TiO_2_ NT array and washed it twice with ethanol and pure water, respectively. The obtained TiO_2_ nanotube@Au nanoparticle (TiO_2_ NT@Au NP) composite was dried in the atmosphere.

In order to prepare molybdenum oxide precursor solution, 10 mL of hydrogen peroxide (30 wt%) was slowly dropped into 0.0083 mol of molybdenum powder in an ice water bath. Subsequently, magnetic stirring was carried out for 4 h to obtain fully reacted molybdenum oxide precursor solution. After that, the molybdenum oxide precursor solution was dropped into 25 mL of 0.6 M thiourea aqueous solution under magnetic stirring. After 60 min of magnetic stirring, a uniform and stable mixed solution was prepared. Next, 0.5 g (0.1 g, 0.3 g) of polyvinylpyrrolidone (PVP, 58,000 g/mol) was dissolved in 5 mL of ultrapure water to prepare the surfactant solution. As-prepared TiO_2_ NT array was placed at the bottom of the 50 mL Teflon-lined stainless steel autoclave, then the surfactant solution and the mixed solution was poured into the autoclave in turn. After sealing, the autoclave was subjected to reaction at 200 ℃ for 24 h. After the hydrothermal reaction, the autoclave was naturally cooled, and then the titanium foil attached with the composite material was picked out and washed with ethanol and ultrapure water for three times, respectively, followed by natural drying in the air atmosphere. In order to obtain a perfectly bonded composite, the dried sample was annealed at 450 ℃ in argon atmosphere, and finally the TiO_2_ nanotube@MoS_2_ nanosheet (TiO_2_ NT@MoS_2_ NS) composite was obtained.

### 2.3. Synthesis of TiO_2_ Nanotube@MoS_2_ Nanosheet@Au Nanoparticle (TiO_2_ NT@MoS_2_ NS@Au NP)

The synthesis method of TiO_2_ NT@MoS_2_ NS@Au NP composite was similar to that of TiO_2_ NT@Au NP, except that the TiO_2_ NT array was replaced by TiO_2_ NT@MoS_2_ NS.

### 2.4. Photocatalytic Degradation Measurement

The photocatalysis degradation of methylene blue (MB, 2.5 mL, 3.5 × 10^−6^ mol/L) using TiO_2_ NT (20 min or 60 min)@Au NP catalysts was tested [2]. The variation in MB concentration with light irradiation (300 W Xe-lamp) time was used to characterize the catalytic effect of TiO_2_ NT@Au NP catalysts.

### 2.5. Electrochemical Evaluation

Electrochemical characteristics of as-prepared samples were explored using CR2016 coin-type half cells. All the samples were directly used as binder-free anodes with the area of 1.5 cm^2^, and the counter electrodes were lithium foils. The binder-free anode and lithium foil were separated via Celgard 2400 separator membrane in cells. A solution of LiPF_6_ (1 M, EC:DMC = 1:1 vol%) was used as cells’ electrolyte. To ensure the adequately diffusion of the electrolyte, all sealed cells need to be stood for 12 h before measuring. The LAND CT2001A was used for the galvanostatic charging/discharging test, within the potential of 0.1–3.0 V. Cyclic voltammograms (CVs) were tested by an electrochemical workstation (CHI 660e) under 0.1 mV/s.

## 3. Results and Discussion

### 3.1. Materials Characterization

The morphology of the as-annealed secondary oxidized TiO_2_ nanotube (TiO_2_ NT) is shown in Appendix A. Typically, a regular network structure is formed on the top of the TiO_2_ nanotube, resulting in its large specific surface area [25]. The top of the TiO_2_ NT array has a double-layer honeycomb structure, which is conducive to photocatalysis and energy storage [25,26,27]. Therefore, we use the secondary oxidized TiO_2_ nanotube (TiO_2_ NT) as a matrix to prepare high-performance catalyst and lithium battery anode. Figure 1a,b shows the SEM images of the TiO_2_ nanotube@Au nanoparticle (TiO_2_ NT@Au NP) composite. The length of the TiO_2_ NTs is around 6 μm, the inner diameter is around 50 nanometers, and the outer diameter is around 140 nm. In addition, the size of Au nanoparticles is between 10 and 25 nm. Au nanoparticles are evenly distributed on the top and outer wall of the TiO_2_ NTs (Appendix A).

The presence of PVP (as a surfactant) is an important factor for the successful preparation of TiO_2_ nanotube@MoS_2_ nanosheet (TiO_2_ NT@MoS_2_ NS) composites. When the surfactant is not present, the MoS_2_ nanosheet cannot be coated on the TiO_2_ nanotube (Appendix A). With the increase in the mass of the surfactant, the MoS_2_ NS coating of the TiO_2_ NT is gradually perfect (Appendix A and Figure 1c,d). While the mass of PVP is 0.5 g, the MoS_2_ NS is well coated on the TiO_2_ NT to form a perfect TiO_2_ NT@MoS_2_ NS composite. Furthermore, the TiO_2_ nanotube@MoS_2_ nanosheet@Au nanoparticle (TiO_2_ NT@MoS_2_ NS@Au NP) composites were also successfully prepared; as shown in Figure 1e,f, the TiO_2_ NT@MoS_2_ NS composites were modified by Au nanoparticles. Thanks to the simultaneous appearance of MoS_2_ nanosheets and Au nanoparticles, the TiO_2_ NT@MoS_2_ NS@Au NP composite has improved lithium capacity and excellent conductivity, which will be shown in the following section.

The structure information of various samples was detected by Raman (Figure 2). The Raman spectrum of TiO_2_ NT shows five peaks, corresponding to E_g_, E_g_, B_1g_, A_1g_, and E_g_ vibration modes of anatase TiO_2_, which is consistent with the literature [28,29,30]. Additionally, the anatase crystal structure of the TiO_2_ nanotube was further confirmed by XRD [31,32,33] and TEM [28,33,34] results (Appendix A). The Raman spectra of TiO_2_ NT@Au NP composite are similar to that of the pristine TiO_2_ NT, because Au (as a metal material) does not show Raman vibration peak. MoS_2_ nanosheet powder’s Raman spectrum shows two peaks at around 379 and 405 cm^−1^, matching the $E_{2g}^{1}$ (in-plane) and A_1g_ (out-of-plane) vibration modes, respectively [35]. The peak position difference of $E_{2g}^{1}$ and A_1g_ vibration modes (Δω) is related to the number of layers of MoS_2_. As the number of layers of MoS_2_ changes from monolayer to bulk phase, the value of Δω increases from 19.57 to 25.5 cm^−1^ [36]. The frequency difference Δω between $E_{2g}^{1}$ and A_1g_ vibration modes of MoS_2_ powder (Appendix A) is 26 cm^−1^, indicating that MoS_2_ powder is a bulk material [35]. The Raman spectrum of TiO_2_ NT@MoS_2_ NS composite not only contains the typical vibration modes of anatase TiO_2_, but also contains the $E_{2g}^{1}$ and A_1g_ vibration modes of MoS_2_. Particularly, the Δω of MoS_2_ coating is 24, in TiO_2_ NT@MoS_2_ NS composite, indicating that the coating is four-layer MoS_2_ [37]. The transition of MoS_2_ from bulk phase of MoS_2_ powder to four layers of MoS_2_ coating may be due to the gap between TiO_2_ NTs limiting the accumulation of MoS_2_, resulting in the formation of four-layer MoS_2_. Similarly, the MoS_2_ coating in the TiO_2_ NT@MoS_2_ NS@Au NP composite is also four-layer MoS_2_. Combined with SEM and Raman results, it can be seen that the TiO_2_ NT@Au NP, TiO_2_ NT@MoS_2_ NS, and TiO_2_ NT@MoS_2_ NS@Au NP composites were successfully prepared.

### 3.2. Photocatalytic Properties of TiO_2_ NT@Au NP Composites

In order to measure the photocatalytic activities of TiO_2_ NT@Au NP composites with a nanotube oxidation time of 20 min (TiO_2_ NT (20 min)@Au NP), the experiment of photocatalytic degradation of methylene blue (MB) was carried out. For comparison, TiO_2_ nanotubes with a secondary oxidation time of 60 min were also used as substrates to successfully prepare TiO_2_ NT (60 min)@Au NP composites, which was also used as a catalyst for degradation of MB. The TiO_2_ NT (60 min) also has a self-organized tube morphology similar to the TiO_2_ NT (20 min) (Figure 1a,b), and the Au particles are evenly anchored on the top and outside of the TiO_2_ NT (60 min) in TiO_2_ NT (60 min)@Au NP composite (Figure 3). Unlike the TiO_2_ NT (20 min)@Au NP composites, in TiO_2_ NT (60 min)@Au NP composites, the length of the nanotubes is 13 μm and the double-layer honeycomb structure on the top of the nanotubes becomes thinner. The thinning of the double-layer honeycomb structure is mainly due to the corrosion of the double-layer honeycomb structure after long-term (60 min) exposure to the electrolyte (containing F^−^) during secondary oxidation process [2,38]. In addition, compared with the TiO_2_ NT (20 min)@Au NP composite, the TiO_2_ NT (60 min)@Au NP composite has more Au particles anchored on the nanotubes, which may be due to more defects on the TiO_2_ nanotubes (60 min) providing more binding sites for the growth of Au nanoparticles [30,39]. The extra defects in TiO_2_ NTs (60 min) are also caused by long-term exposure of NTs to corrosive electrolyte (containing F^−^) [2,38].

Figure 4a shows the degradation curve of MB using different samples as catalysts under UV-visible light irradiation. There was no catalyst in the control experiment, and only UV-visible light irradiation was carried out. The control experiment exhibits only a small amount of degradation of MB, which may be due to thermal degradation caused by UV-visible light irradiation. The TiO_2_ NT (20 min) shows no photocatalytic effect, due to the high electron-hole pair recombination rate. Compared with TiO_2_ NT (20 min), the catalytic effect of TiO_2_ NT (60 min) is improved due to the increase in the amount of catalyst and defects. Abundant defects may induce the generation of defect energy levels in the TiO_2_’s band gap. The appearance of defect energy level increases the diffusion length of carriers, prolongs the life of carriers, hinders the recombination of electron/hole, improves the utilization of light, and increases the catalytic effect [40]. The presence of Au particles acted as separation centers in the TiO_2_ NT (20 min)@Au NP composite, which reduces the chance of electron-hole pair recombination [11], thus enhancing the photocatalytic effect. Therefore, the TiO_2_ NT (20 min)@Au NP composite has remarkable photocatalytic properties (Figure 4). Previous studies showed that the photocatalytic properties of the TiO_2_ matrix composites loaded with Au particles are related to the size [11,41,42] and the density [41,43,44] of Au particles. When the size of nanoparticles is smaller than 5 nm, the catalytic effect of TiO_2_ matrix composites is more effective [41,43,44]. While the content of Au is 2%, TiO_2_ matrix composite has the optimal photocatalytic performance [11,41,42]; meanwhile, if the Au content is too large, it will be harmful to the photocatalytic effect [11,41]. Particularly, Figure 4 exhibits that although the TiO_2_ NT (60 min) shows higher photocatalysis activity than the TiO_2_ NT (20 min), the photocatalysis activity of TiO_2_ NT (60 min)@Au NP composite is worse than that of the TiO_2_ NT (20 min)@Au NP. In TiO_2_ NT (20 min)@Au NP and TiO_2_ NT (60 min)@Au NP composites, the size of Au particles is similar, but the content of Au is significantly different (Figure 1a,b and Figure 3). The poorer photocatalytic activity of TiO_2_ NT (60 min)@Au NP is mainly due to excessive Au content. When the Au content exceeds the optimum, the Au particles act as the recombination center of the electron-hole pair, which impairs the catalytic effect [11,41].

The schematic diagram of the photocatalytic degradation of MB by TiO_2_ NT (20 min)@Au NP composites is illustrated in Figure 4b. Titanium oxide (TiO_2_ NT), as an *n*-type semiconductor material, has a work function of 4.2 eV and a band gap of 3.2 eV [45,46], and gold has a work function of 5.0 eV [47]. Thus, a Schottky junction is formed at the Au NP/TiO_2_ NT interface, because the gold’s work function is larger than that of *n*-type TiO_2_. The appearance of Schottky junction increases the separation of electron-hole pairs and improves the photocatalysis degradation activity of TiO_2_ nanotube [11,42]. Specifically, under the irradiation of UV-visible light, the electrons in TiO_2_ NT are excited to the conduction band from the valence band, while leaving holes with positive charges in the valence band (Figure 4b). These photogenerated electrons are transferred to Au particles, activating the adsorbed oxygen molecules (O_2_) into superoxide radicals (·O_2_^−^) [11,47]. Holes are left on the valence band of TiO_2_ NT for the surface oxidation reaction to generate hydroxyl radicals (·OH) [12]. Finally, the ·O_2_^−^ and ·OH can react with MB to produce inorganic substance (e.g., CO_2_ and H_2_O) [48]. Herein, the high electron separation and transfer ability at the TiO_2_/Au interface inhibits the high electron-hole recombination of TiO_2_ NT (20 min), which explains the improvement of the photocatalysis efficiency of the TiO_2_ NT (20 min)@Au NP composites. However, the content of gold is too large in the TiO_2_ NT (60 min)@Au NP composite, thus the Au particles with a lot of negative charges become the hole capture center, increasing the electron-hole recombination, and damaging the photocatalytic efficiency [11,42,49]. So far, the TiO_2_ NT (20 min)@Au NP composite shows excellent photocatalytic properties because of its large specific surface area and appropriate Au content, and this material with enhanced conductivity will also show good application potential in lithium ion energy storage.

### 3.3. Electrochemical Measurements

Anatase TiO_2_ nanotube with a duration of 20 min of secondary anodic oxidation (named TiO_2_ NT) was selected as the matrix for the preparation of composite anodes for lithium battery, due to the perfect double-layer honeycomb surface and large specific surface area of the nanotubes. The conductivity of the TiO_2_ NT@Au NP composite is analyzed and predicted by the energy band diagram shown in Figure 5. The Schottky junction was formed in the TiO_2_ NT@Au NP composite (Figure 5b), and the Schottky barrier is 1.0 eV (eϕ_Bn_ = 1.0 eV), and the built-in electric field barrier is 0.8 eV (eV_bi_ = 0.8 eV). When we apply a positive voltage to TiO_2_ relative to gold (Figure 5c), the Schottky junction is reverse biased. A large number of electrons can easily cross the Schottky barrier from Au to TiO_2_, because the Schottky barrier (eϕ_Bn_) remains unchanged. Meanwhile, if we apply a positive voltage to gold relative to TiO_2_ (Figure 5d), the TiO_2_ NT/Au NP junction is forward biased. In the case of positive bias of Schottky junction, the electrons can easily pass through the entire TiO_2_ NT@Au NP composite by overcoming a reduced potential barrier e(V_bi_ − V) (Figure 5d). The TiO_2_ NT@Au NP composite exhibits good electron flow characteristics in two kinds of electric fields with opposite directions (responding to charging/discharging electric field of lithium battery), which reveals that the appearance of Schottky junction improves the conductivity of TiO_2_ nanotubes. Enhanced conductivity will improve the electrochemical performance of TiO_2_ NT@Au NP anode, which will be revealed in the subsequent electrochemical analysis.

The energy band diagram of TiO_2_ NT@MoS_2_ NS and TiO_2_ NT@MoS_2_ NS@Au NP heterojunctions is shown in Appendix A. Appendix A shows the energy band diagrams of anatase TiO_2_ NT, MoS_2_ NS [50,51,52,53], and Au NP before contact. The nn isotype heterojunction was formed in the TiO_2_ NT@MoS_2_ NS composite (Appendix A). If a reverse-biased voltage (V, positive voltage to TiO_2_ relative to MoS_2_) is applied across the heterojunction, the built-in electric field barrier eV_bi_ increases to e(V_bi_ + V) (Appendix A). Similarly, if a forward bias is applied, the eV_bi_ is reduced to e(V_bi_ − V). Under the presence of nn isotype heterojunction, electrons can easily flow through the entire composite material in the opposite electric field direction (responding to charging/discharging electric field of lithium battery), as shown in Appendix A. Appendix A shows the energy band diagram of TiO_2_ NT@MoS_2_ NS@Au NP heterojunction accompanied by the anatase TiO_2_/MoS_2_ interface (nn isotype heterojunction) and the MoS_2_/Au interface (Schottky heterojunction). Based on the above analysis of Schottky and nn heterojunctions, electrons also can easily flow through the entire TiO_2_ NT@MoS_2_ NS@Au NP electrode in charge/discharge electric fields, resulting in good conductivity of the electrode. Therefore, all as-prepared composites have enhanced electrical conductivity, which is conducive to improving the electrochemical performance of them.

Galvanostatic charge–discharge curves of the as-prepared anodes are displayed in Figure 6. TiO_2_ NT has a typical charging/discharging voltage plateau at 2.0/1.7 V (Figure 6a), which corresponds to the lithium extraction from and insertion into the anatase phase TiO_2_, respectively [54,55]. The initial discharge capacity and initial coulomb efficiency of the TiO_2_ NT are 188.5 μAh cm^−2^ and 60.0% (Table 1). Figure 6b demonstrates the charge–discharge curves of the TiO_2_ NT@Au NP composite anode, obviously, the initial discharge capacity (310.6 μAh cm^−2^, 1.6 times of the pristine TiO_2_ NT) and initial coulomb efficiency (76.5%) of the composite were more significantly improved than pristine TiO_2_ NT. Au particles can store lithium-ion in the form of alloy [55]. The modification of Au particles not only improves the lithium storage capacity of NTs, but also markedly improves the conductivity of TiO_2_ NTs (Figure 5) enhancing electrochemical lithium storage performance. The TiO_2_ NT@MoS_2_ NS composite anode exhibits an initial discharge capacity of 391.3 μAh cm^−2^ and an initial coulomb efficiency of 51.0% (Table 1). Due to the introduction of MoS_2_, the capacity of the TiO_2_ NT@MoS_2_ NS composite was significantly improved (2.1 times the capacity of the pristine TiO_2_ NT), and the charge/discharge voltage plateaus moved to 2.1/1.65 V [53]. Because the TiO_2_ NT@MoS_2_ NS composite underwent 450 ℃ high-temperature annealing in argon atmosphere, there are abundant defects in the nanotubes of the composite anode, so the charge–discharge curve is more inclined with smaller voltage plateaus, indicating a more amorphous crystal structure and a more uniform lithium intercalation process [54,56,57]. Due to the introduction of defects, the conductivity of TiO_2_ NT@MoS_2_ NS was also improved, which is conducive to the improvement of its electrochemical performance [28,58]. However, the low initial coulomb efficiency (51.0%) of the TiO_2_ NT@MoS_2_ NS corresponds to a large irreversible capacity, which is mainly caused by the irreversible lithium insertion of MoS_2_ [59,60]. To further enhance the electrochemical properties of the TiO_2_ NT@MoS_2_ NS, we modified the TiO_2_ NT@MoS_2_ NS composite with Au nanoparticles. The TiO_2_ NT@MoS_2_ NS@ Au NP composite contains not only MoS_2_ with high capacity, but also Au particles with high conductivity, so the composite material has large initial discharge-specific capacity (487.4 μAh cm^−2^) and high initial coulomb efficiency (65.8%).

Figure 7a shows the third scan cycle CV curves of four different samples under a scanning rate of 0.1 mV/s. Obviously, the electrochemical behaviors of the samples revealed by CV curves is similar to that revealed by the galvanostatic charge–discharge curves (Figure 6). A pair of obvious redox peaks (around 2.5/1.3 V) responds to the process of lithium extraction from and insertion into anatase TiO_2_ [22,54]. Consistent with the literature, the addition of gold does not cause additional peaks in the CV curves [61,62]. In addition, there is no typical peak of MoS_2_ because of the low content of molybdenum sulfide in the composites [22], as well as the overlap of the peak positions of MoS_2_ and TiO_2_ [22,30,53,60] forming broad peaks. In particular, the anodic peak and cathodic peak of the TiO_2_ NT@MoS_2_ NS composite moved to 2.7 and 1.2 V, respectively, corresponding to the shift of the voltage plateaus in the galvanostatic charge–discharge curve (Figure 6c). Such a shift in these peak positions is mainly due to the introduction of molybdenum sulfide in the TiO_2_ NT@MoS_2_ NS composite anode [53]. Another point to note is that the area of CV curves is proportional to the lithium storage capacity of composite anodes. The value of the CV area of the as-prepared samples has the same order as the value of the specific capacity revealed by the galvanostatic charge–discharge curves (TiO_2_ NT < TiO_2_ NT@MoS_2_ NS < TiO_2_ NT@Au NP < TiO_2_ NT@MoS_2_ NS@Au NP).

Cyclic stability is an important part of the performance of lithium batteries in practical applications. The cyclic stability of the TiO_2_ NT@MoS_2_ NS@Au NP anode is shown in Figure 7b. The well-designed TiO_2_ NT@MoS_2_ NS@Au NP anode shows the highest specific capacity; even after 50 cycles, it still has a specific capacity of up to 273.6 μAh cm^−2^. After 50 cycles, TiO_2_ NT@MoS_2_ NS@ Au NP has a capacity retention rate of 81.0% (compared with the second discharge capacity), which is higher than that of TiO_2_ NT@Au NP (67.8%) and TiO_2_ NT@MoS_2_ NS (74.8%) anodes (Table 1). The high capacity and excellent stability of the TiO_2_ NT@MoS_2_ NS@Au NP composite are mainly attributed to several factors. First, the small volume expansion coefficient of the TiO_2_ NT matrix during the charge–discharge cycle leads to excellent cycle stability. Second, the high theoretical specific capacity of MoS_2_ improves the capacity of the composite. Third, the introduction of defects and Au particles makes TiO_2_ NT@MoS_2_ NS@Au NP have excellent electronic/ion conductivity. The as-designed TiO_2_ NT@MoS_2_ NS@Au NP electrode will show great potential in practical applications of the lithium-ion battery, and the design concept of the composite also has an important inspiration for the design of other ideal lithium-ion battery anode.

The button batteries were disassembled in glove box after 50 lithiation/delithiation cycles. SEM images of cycled anode materials are shown in Appendix A. The TiO_2_ nanotube anode maintains the best tubular morphology (Appendix A), without significant expansion and breakage [2,17], corresponding to the highest cycle stability (Table 1). In TiO_2_ NT@Au NP composite, Au particles fall off and agglomerate during the cycling process, resulting in a significant capacity degeneration with the lowest capacity retention (Table 1). Due to the inherent low structural strength of MoS_2_ [17,59], TiO_2_ NT@MoS_2_ NS composite’s morphology underwent significant changes, accompanied by the crushing of MoS_2_ nanosheets and the fracture of TiO_2_ nanotubes (Appendix A). In the TiO_2_ NT@MoS_2_ NS@Au NP composite, the collapse of the thin MoS_2_ NS coating is mitigated by highly stable TiO_2_ nanotubes [17], so the MoS_2_ NS in the composite is not subject to pulverization. In addition, the presence of MoS_2_ nanosheets stabilizes the attachment of gold nanoparticles, and the gold particles do not fall off after cycling. Therefore, TiO_2_ NT@MoS_2_ NS@Au NP composite has improved cycle stability compared to TiO_2_ NT@Au NP and TiO_2_ NT@MoS_2_ NS composites (Table 1).

## 4. Conclusions

The carefully designed ternary TiO_2_ nanotube@MoS_2_ nanosheet@Au nanoparticle (TiO_2_ NT@MoS_2_ NS@Au NP) composite with excellent electrochemical performance was successfully prepared via combining two-step electrochemical anodization and the hydrothermal method. The TiO_2_ NT@MoS_2_ NS@Au NP anode with abundant defects and highly ordered arrangement demonstrates outstanding structural stability after charge/discharge cycles. Therefore, the TiO_2_ NT@MoS_2_ NS@Au NP anode exhibits not only high discharge capacity (487.4 μAh cm^−2^), but also excellent capacity stability (a capacity retention rate of 81.0%, after 50 cycles). In addition, as a multifunctional material, the TiO_2_ nanotube@Au nanoparticle (TiO_2_ NT@Au NP) composite showed excellent photocatalytic degradation performance and enhanced electrochemical performance. The excellent performances of multi-functional TiO_2_ NT@Au NP composites can be attributed to following accounts: (1) the double-layer honeycomb surface structure of TiO_2_ NT matrix makes the composite have a large surface area. (2) The appropriate concentration of gold as the separation center reduces the recombination of electron-hole pairs. (3) The excellent conductivity of Au/TiO_2_ NT Schottky junction improves the electron and ion transport efficiency. Therefore, TiO_2_ NT matrix composites, including TiO_2_ NT@Au NP, TiO_2_ NT@MoS_2_ NS, and TiO_2_ NT@MoS_2_ NS@Au NP, exhibit excellent potential in photocatalysis and lithium storage, which will open a new avenue for pollutant degradation and energy storage.

## Figures and Tables

**Figure 1 materials-16-02716-f001:**
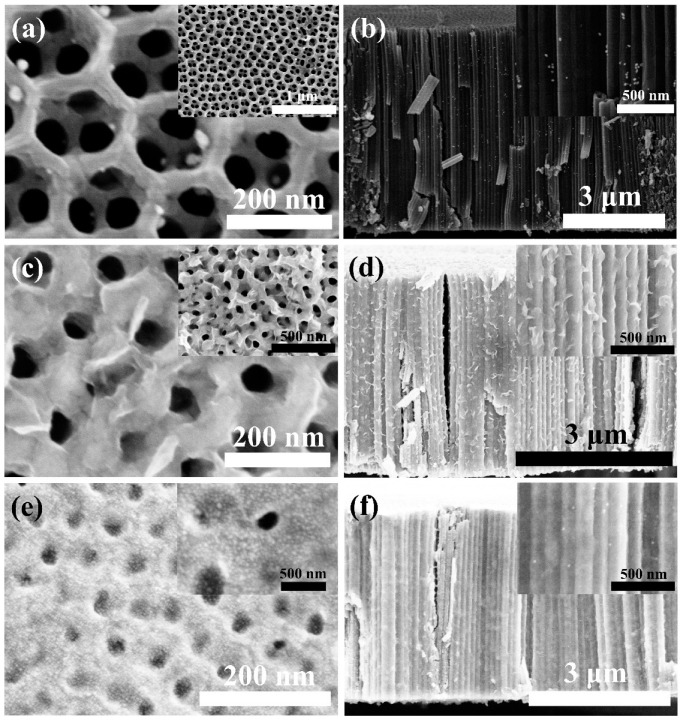
SEM images of (**a**,**b**) TiO_2_ nanotube@Au nanoparticle (TiO_2_ NT@Au NP) composite, (**c**,**d**) TiO_2_ nanotube@MoS_2_ nanosheet (TiO_2_ NT@MoS_2_ NS, 0.5 g PVP) composite, and (**e**,**f**) TiO_2_ nanotube@MoS_2_ nanosheet@Au nanoparticle (TiO_2_ NT@MoS_2_ NS@Au NP) composite.

**Figure 2 materials-16-02716-f002:**
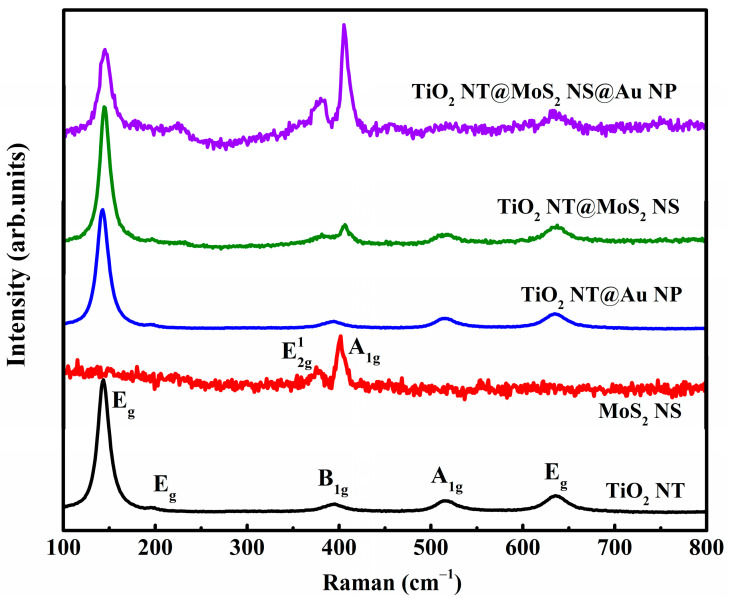
Raman spectra of TiO_2_ NT, MoS_2_ NS, TiO_2_ NT@Au NP, TiO_2_ NT@MoS_2_ NS, and TiO_2_ NT@MoS_2_ NS@Au NP composites.

**Figure 3 materials-16-02716-f003:**
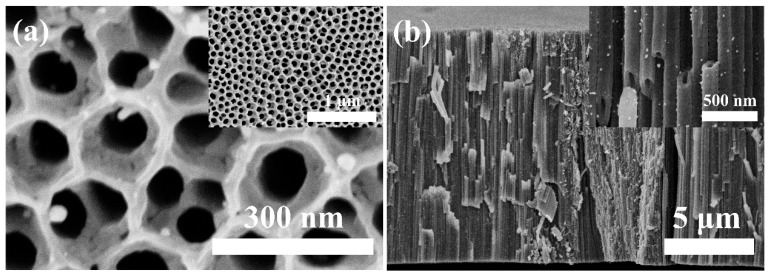
SEM images of TiO_2_ NT@Au NP composite with 60 min growth time of TiO_2_ nanotube. (**a**) Top view and (**b**) cross-section view SEM images of the TiO_2_ NT (60 min)@Au NP composite.

**Figure 4 materials-16-02716-f004:**
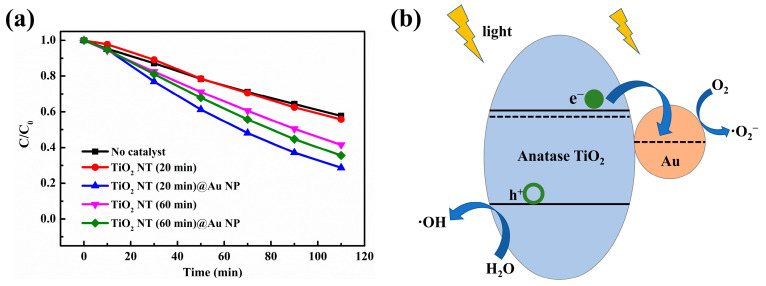
(**a**) TiO_2_ NT and TiO_2_ NT@Au NP composites with different NT growth times (20 min and 60 min), are used as catalysts for photocatalytic degradation of MB. (**b**) Schematic diagram of photocatalytic degradation of MB by TiO_2_ NT (20 min)@Au NP composite.

**Figure 5 materials-16-02716-f005:**
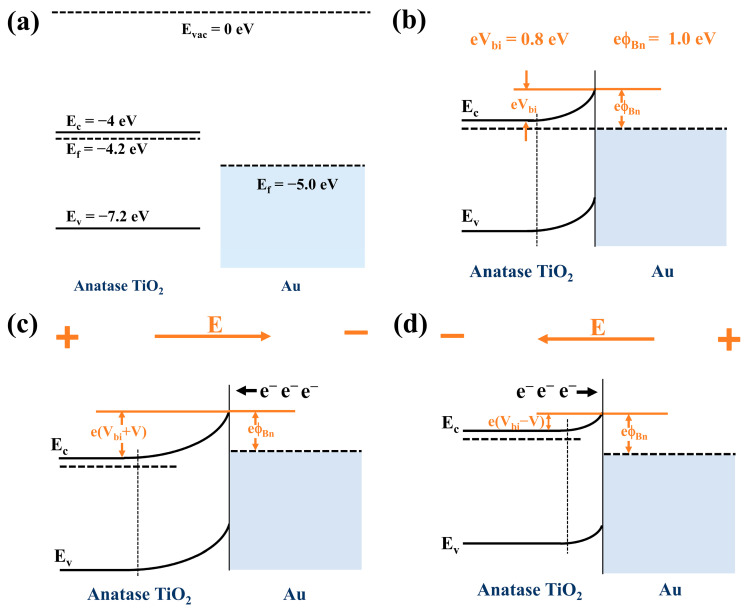
Energy band diagrams for TiO_2_ NT@Au NP (**a**) before contact, and (**b**) after contact for thermal equilibrium. (**c**) The positive voltage V is applied to the TiO_2_ NT relative to the Au NP, and (**d**) the positive voltage V is applied to Au NP relative to the TiO_2_ NT.

**Figure 6 materials-16-02716-f006:**
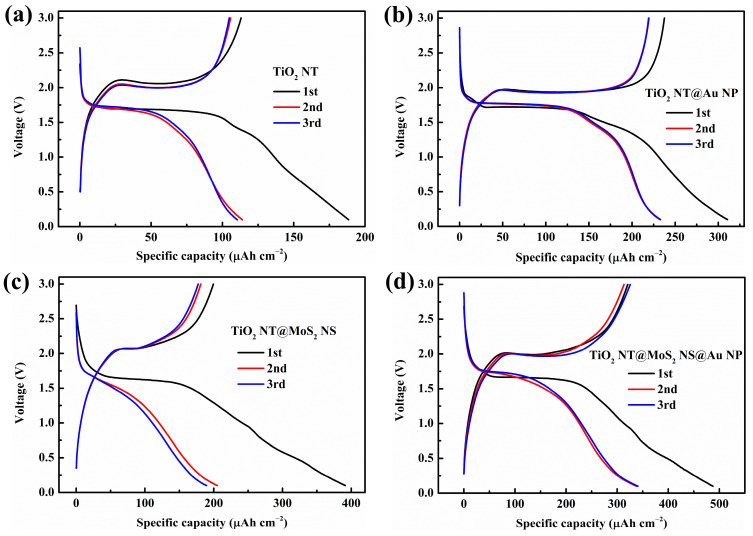
The initial three charge/discharge profiles of (**a**) TiO_2_ NT, (**b**) TiO_2_ NT@Au NP, (**c**) TiO_2_ NT@MoS_2_ NS, and (**d**) TiO_2_ NT@MoS_2_ NS@Au NP composites, under 100 μA cm^−2^.

**Figure 7 materials-16-02716-f007:**
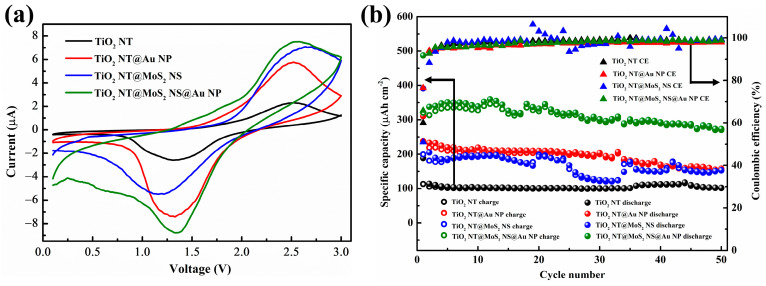
(**a**) Third cycle CV curves, as well as (**b**) the cycling stability and coulomb efficiency of TiO_2_ NT, TiO_2_ NT@Au NP, TiO_2_ NT@MoS_2_ NS, and TiO_2_ NT@MoS_2_ NS@Au NP anodes.

**Table 1 materials-16-02716-t001:** Initial discharge capacity, initial coulomb efficiency, and capacity retention rate after 50 cycles (compared with the second cycle) of various as-prepared samples.

	Capacity	Initial Discharge Capacity (μAh cm^−2^)	Initial Coulomb Efficiency (%)	Capacity Retention after 50 Cycles (%)
Samples				
TiO_2_ NT		188.5	60.0	89.6
TiO_2_ NT@Au NP		310.6	76.5	67.8
TiO_2_ NT@MoS_2_ NS		391.3	51.0	74.8
TiO_2_ NT@MoS_2_ NS@Au NP		487.4	65.8	81.0

## Data Availability

The data presented in this work are available from the corresponding authors upon request.

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
