# Peer review of "Multifunctional TiO2 Nanotube-Matrix Composites with Enhanced Photocatalysis and Lithium-Ion Storage Performances"

_materials, 2023, doi:10.3390/ma16072716_

Round 1

Reviewer 1 Report

Thank you for invite me as reviewer with title Multifunctional TiO2 Nanotube-Matrix Composites with Enhanced Photocatalysis and Lithium-lon Storage Performances. I have opinion and questions about some part of this paper need clarification.

1.       Line 7 please make sure the correspondence (double type)

2.       From (line 68-74) synthesis of TiO2 nanotube author use 2-time electrochemical anodization, what the reason? As my experience we can fine directly fine TiO2 nanotube just one-time anodization.

3.       Line 81-82, author head till boiled, what temperature needed?

4.       This paper just mentions about TiO2 anatase type, which kind author that claim this Anatase phase, just used XRD result?

5.       If we use TiO2, normally in this type we find bandgap 3.2 – 3.3 eV than we add material like Au it’s fine from optical view, but this paper also using MoS but not mention about banding from both materials like with Au in junction. Can you make explain, please?

6.       Figure 7, we can see 4 CV curve, but author mention 3 CV curve?

7.       Still in CV Curve, if we using several materials, however with differ time deposition, we can fine some peak during CV measurement, but here, we just see one peak in redox peak. What the reason?

8.       Figure 7B author can add symbol to guide which line to Specific capacity and Coulomb efffeciency.

9.       For SI can you add SEM image after all assessment, we can know about the degradation after view time.

Overall this paper quite good and we can except after several explanation and responses.

Thanks’ you

Author Response

Dear Reviewer,

Reviewer 2 Report

see comments

Author Response

Dear reviewer,

Reviewer 3 Report

In this article , the authors report about application of TiO2. Herein, TiO2 nanotube (TiO2 NT) 10

with a double-layer honeycomb structure were prepared by electrochemical anodization. And the unique TiO2 NT were used as matrixes to prepare multifunctional composites.

The article is interesting. However, some issues should be addressed.

1)      In the introduction, the authors should clearly highlight the novelty and the importance  of their work in comparison to previous works. 

         2)  The conclusion section should be extended and improved to help a reader not familiar with the topic.

3) In the abstract the authors should highlight the novelty of their work .  

4) In the introduction the authors should refer to recent advances about TiO2 nanotubes and MoS2.

See for example:

·       Influencing parameters in the electrochemical anodization of TiO2 nanotubes: Systematic review and meta-analysis,Ceramics International, Volume 48, Issue 14, 2022, Pages 19513-19526, ISSN 0272-8842, https://doi.org/10.1016/j.ceramint.2022.04.059.

·       Variable angle spectroscopic ellipsometry characterization of spin-coated MoS2 films, Vacuum,

Volume 189, 2021, 110232, ISSN 0042-207X,  https://doi.org/10.1016/j.vacuum.2021.110232. (https://www.sciencedirect.com/science/article/pii/S0042207X21001871)

·       Synthesis of Cross-like TiO2 Thermally Derived from Ammonium Oxofluorotitanate Mesocrystals Under Different Calcination Temperatures and Their Photocatalytic Activity. Electronic Materials Letters 225.

5)      In Figure 2 the authors should use arb.units instead of a.u.

Author Response

Dear Reviewer,

Round 2

Reviewer 3 Report

I recommend the publication of this article.